# The Gut‒Breast Axis: Programming Health for Life

**DOI:** 10.3390/nu13020606

**Published:** 2021-02-12

**Authors:** Juan M. Rodríguez, Leónides Fernández, Valerie Verhasselt

**Affiliations:** 1Department of Nutrition and Food Science, Complutense University of Madrid, 28040 Madrid, Spain; 2Department of Galenic Pharmacy and Food Technology, Complutense University of Madrid, 28040 Madrid, Spain; leonides@ucm.es; 3School of Molecular Sciences, University of Western Australia, M310 Perth, WA 6009, Australia; valerie.verhasselt@uwa.edu.au

**Keywords:** human milk, gut‒breast axis, entero-mammary pathway, immunomodulation, tolerance, gut colonization

## Abstract

The gut is a pivotal organ in health and disease. The events that take place in the gut during early life contribute to the programming, shaping and tuning of distant organs, having lifelong consequences. In this context, the maternal gut plays a quintessence in programming the mammary gland to face the nutritional, microbiological, immunological, and neuroendocrine requirements of the growing infant. Subsequently, human colostrum and milk provides the infant with an impressive array of nutrients and bioactive components, including microbes, immune cells, and stem cells. Therefore, the axis linking the maternal gut, the breast, and the infant gut seems crucial for a correct infant growth and development. The aim of this article is not to perform a systematic review of the human milk components but to provide an insight of their extremely complex interactions, which render human milk a unique functional food and explain why this biological fluid still truly remains as a scientific enigma.

## 1. Introduction

The intestine could be defined as the “ugly duckling” of our body; it has provided so many surprises to the scientific community in the last decades that it currently appears to us as a truly fascinating organ. First of all, large volumes of food, chemicals and microorganisms pass through this huge organ continuously, some of them potentially toxic or pathogenic. On the other hand, it is in charge of carrying out a remarkably efficient rescue operation: the rescue of the essential nutrients existing in an intermittent magma of food and water. To achieve this, it has to conduct this content at an adequate speed; provide various secretions; perform digestion tasks; absorb the digested products, electrolytes, and water; send this material into the circulatory stream; and, finally, expel waste products. 

As if this were not enough, it is a keystone of the immune system, performing a conscientious task of sampling the enteric environment and having to maintain a difficult balance: firmly controlling pathogens, tolerating diners, and originating proportionate responses in front of the crowd of antigens with which it has contact. There is more still: the gut is the host of a certainly complex microbiota, which, directly or indirectly, contributes to most, if not all, host functions. Unveiling the roles of the gut microbiota in health and disease remains one of the most important challenges in biomedicine, and this may provide a key knowledge for medical personalization. Finally, the intestine is a complex neuroendocrine organ. It maintains a constant exchange of information with the brain and with other organs through a complex network of nervous, metabolic, and endocrine signals. The enteric nervous system is a vast chemical warehouse in which each of the neurotransmitters that operate in our brain is represented. Its mission goes far beyond the supervision of the already complex digestive processes and includes the production of psychoactive substances. The enteric nervous system is a place of neural integration and processing, which is why it has been considered our “second brain” [1]. It is illustrative that the intestine is precisely the organ in which the physical connections between the immune system and the nervous system were first revealed [2].

As a result, the gut is in the crossroads of health and disease, programming and modulating the function of distant organs. In fact, in the last decades, we have assisted in the blooming of an increasing number of axes names used to highlight the functional connections of the gut with almost any other human organ: gut‒brain axis [3], gut‒lung axis [4], gut‒liver axis [5], gut‒kidney axis [6], gut‒skin axis [7], gut‒joint axis [8], gut‒bone axis [9], gut‒pancreas axis [10], gut‒heart axis [11], gut‒spleen axis [12], gut‒vagina axis [13], and gut‒eye axis [14]. Among this myriad of interconnected axes, one of the most relevant ones for our biological history is usually forgotten: the gut‒breast axis.

## 2. The Maternal Gut‒Breast Axis

### 2.1. Gut and Breast Adaptations for Pregnancy and Lactation

The maternal gut plays a key role in programming and training the mammary gland to meet the nutritional, microbiological, immunological, and neuroendocrine requirements of the growing infant [15] and, therefore, in the short- and long-term benefits associated to breastfeeding [16,17]. Pregnancy and lactation require extensive changes in a woman’s body, involving most, if not all, organs and systems. Among them, the morphological and physiological adaptations of the maternal gastrointestinal tract are particularly relevant for the success of these two key life stages. These adaptations affect absorption ability, metabolism (including bacterial metabolism), and the composition of both the microbiota and the immune system [18,19,20,21,22]. Overall, the volume of the mucosal surface of the gut increases sharply to create an enormous absorptive area, especially in those parts (duodenum and ileum) that are highly involved in the breakdown of food components and nutrient absorption [19,23]. In addition, the conditions created during the last trimester of pregnancy favors a physiological translocation of immune and bacterial cells [24]. 

Parallel, there is a deep structural and functional remodeling of the mammary gland, including a huge process of angiogenesis, leading to the development of a highly specialized exocrine organ, ready for the transmission of a vast array of nutrients and bioactive compounds and cells to the infant gut. Gut and mammary adaptations are perfectly synchronized, since both organs establish complex and complementary interactions [25]. As stated by Weaver [18], “the relationship between lactating mammary function and neonatal gastrointestinal function is an example of the parallel evolution of two organs, the breast and gut, which after birth, together undertake functions performed by the placenta during intrauterine life.”

### 2.2. Maternal Gut and Breast: The Nutritional Connection 

The main sources of milk components are maternal diet, maternal stores, and de novo biosynthesis by the lactocytes or by other cells (immune cells, microbes) inhabiting the mammary epithelium and/or carried by milk [26] (Figure 1). In this context, a correct functionality of the gastrointestinal tract (absorption, metabolism) is strictly necessary to assemble the components of human milk, either directly from the maternal diet or to provide the immediate principles (fatty acids, amino acids, glucose), water, and electrolytes required for the maternal cells to build them up (Figure 2). 

The nutritional composition of human milk is quite conserved, but there are some nutrients (thiamine, riboflavin, vitamin B-6, vitamin B-12, choline, retinol, vitamin A, vitamin D, selenium, iodine, and fatty acids, including monounsaturated, n–6, and n–3 fatty acids) that are critical for infant development and health, and whose concentrations are highly affected by maternal diet or nutritional status [27,28,29,30,31,32]. In fact, the presence of these nutrients in milk is rapidly and/or substantially reduced by maternal deficit or depletion, and this situation has a negative impact on infant health and development. The concentrations of many of such nutrients are often below recommended levels in many populations. This situation is particularly worrying in low income countries, but it can be also observed in developed countries (e.g., omega-3 fatty acids, choline) [27,28,29,32,33,34]. This does not mean that breastfeeding is not the best feeding practice in these settings, but that the diet of the mother may require supplementation with some essential nutrients when deficiencies are observed. The concentrations of such nutrients can be restored following maternal supplementation, a practice that will benefit both maternal and infant health. This is in contrast with other nutrients, such as lactose, protein, calcium, or folate, which milk concentrations remain relatively largely unaffected by maternal diet [32]. However, in this case diet deficiencies will affect maternal health and, in the long term, they will indirectly affect infant’s health too. Unfortunately, research on how maternal diet affects the micro- and macronutrients’ content of human milk, both quantitatively and qualitatively, is surprisingly scarce. In this context, conflicting data have been obtained in relation to some minerals, such as iron or zinc, which are essential for a wide array of physiological functions [35,36], and which deficiencies in the mother‒infant dyad are particularly concerning [37,38]. Some studies have reported that their milk concentrations were associated with maternal dietary intake and/or supplementation, others have found no relationships with these factors and, finally, others have provided ambiguous results [39,40,41,42,43]. More recently, a positive correlation between maternal diet and the milk concentration of both minerals was detected only when the total intake (including diet and supplementation) was taken into account [44]. This and other studies have found that milk levels of these two minerals could be below the adequate intake for infants in a relatively high percentage of samples [34,35,36,37,38,39,40,41,42,43,44]. Positive associations between dietary iron and zinc intakes and their respective serum levels among premenopausal women have been described [45]. However, contrary to iron, zinc concentrations are higher in human milk than in the serum of a breastfeeding woman [46], and, therefore, it is possible that maternal factors may influence the concentration of these two minerals in a different way [44]. Iron and zinc are good examples to highlight that large scale studies are required to provide accurate information about levels, bioavailable forms, and critical factors that are associated with the concentrations of micronutrients in human milk. 

Among human milk nutrients, fatty acids are extremely sensible to maternal nutrition [47]. This is a very relevant issue because of the critical roles of fatty acids for growth (energy source), cardiovascular health, and organ development, playing a keystone role in the composition of the cell membranes, neurodevelopment, immunomodulation, and as modulators of the milk microbiota [29,48,49]. There is evidence showing that, overall, the profile of fatty acids in human milk has changed with modern dietary practices and, at least in some cases, this may lead to profiles that are not adequate for an optimal neurological development of the infant [29]. In addition, intra-individual and inter-individual differences in the diet may account for differences in the composition and structure of milk fat and, as a result, in the concentration of the many fat-associated small molecules [50], and in the structure and composition of fat globules and their membranes, which represent complex colloidal assemblies with plenty of bioactive molecules that may be very relevant for infant outcomes [51]. The structure of the milk fat globule membrane is essential for correct delivery of nutritional and functional lipids and proteins to the infant gut [52] and, as a consequence, for many biological processes relevant for infant health [49,51,53]. This must be taken into account in Human Milk Banks, since processing of donor milk (e.g., pasteurization, freeze/unthaw) may disrupt the structure of the milk fat globules and their membranes and lead to relevant losses in the biological functions.

Oxidative stress is associated with respiratory and intestinal diseases in infants, especially among preterm ones [54]. Interestingly, human milk contains a wide spectrum of endogenous antioxidant compounds, including superoxide dismutase, glutathione peroxidase, catalase, glutathione, or melatonin [55]. The concentration of these antioxidants in human milk is adapted to gestational age, providing higher levels to infants with lower degree of maturation [56]. As a consequence, the level of oxidative stress in premature neonates fed with breast milk is lower than that observed among formula-fed ones [57,58]. Phytochemicals (polyphenols, carotenoids) in the maternal diet provides an exogenous source of antioxidants, since they are also transferred to milk [59]. Although many of the human milk phytochemicals have a wide range of biological activities [60,61,62], most of the research interest has been focused on polyphenols because of their potential as antioxidants. Through their antioxidant activities, diet-related human milk polyphenols might contribute to the benefits that human milk provides to this infant population [63]. Many different flavonoids (epicatechin, epicatechin gallate, epigallocatechin gallate, naringenin, kaempferol, hesperetin, quercetin) can be present, simultaneously, in the same human milk sample [64]. The flavonoid profile depends almost exclusively on maternal diet and rapid shifts can be observed depending on the intake of polyphenol-rich foods (berries, soya, dark chocolate, olive oil, wine) [59]. Polyphenol and other antioxidants are becoming increasingly popular as food supplements among the general population, but their actual impact in human milk composition and in infant outcomes is rather speculative and far from being fully elucidated because of the lack of well-designed epidemiological studies and human clinical trials.

The relevance of maternal diet for infant health has been highlighted by the use of murine models, since pups cross-fostered to dams that received a high-fat diet during lactation develop insulin resistance [65], obesity [66], and hypertension [67], which are symptoms typically related to metabolic syndrome. Metabolic diseases are associated with significant shifts in the gut microbiome [68]. In turn, the maternal gut microbiota can metabolize many food components. Some of the resulting metabolites, such as short chain fatty acids (SCFAs), will be absorbed and transferred to the milk [69]. It must be highlighted that there is an intense traffic of bacterial metabolites from the maternal gut to the breast or to maternal tissues (from which they can later reach the breast) through the bloodstream during late pregnancy and throughout the lactation period [70]. Many of these metabolites, including retinoid compounds, fatty acids, or aryl hydrocarbon receptor ligands, are transferred to the infant gut through breastfeeding and play relevant roles for immune development and infant health (including the metabolism of xenobiotic chemicals) [70,71]. At present, 8.3% of the human milk metabolites included in the Human Metabolome Database (http://www.hmdb.ca) are exclusively produced by certain microbes and, most probably, their presence is the result of the activity of the maternal gut microbiota.

### 2.3. Is There an Entero-Mammary Circulation of Immune and Microbial Cells?

Lactating mammary glands are a part of the maternal‒mucosal immune system [72] and, as such, their immune status has relevant implications for infant development and health [73]. In addition to a wide spectrum of soluble immune factors (which will be discussed later), human milk also contains a variety of leukocytes, including neutrophils, macrophages, and lymphocytes [74]. Although they may transmigrate to infants and seem to play a key role in infant defense and immune training during early life [75], their exact origin and functions remain largely unknown. Interestingly, other types of leukocytes (basophils, eosinophils, mast cells) and platelets, which are usually related to inflammation, are rarely found [74]. The concentration of total leukocytes is higher in colostrum than in mature milk, probably reflecting the fact that tight junctions of the mammary epithelium are looser during the first days after birth [76]. 

Some decades ago, flow cytometry studies showed that the proportions of neutrophils, macrophages, and lymphocytes in human milk are approximately 80%, 15%, and 4%, respectively [77,78]. However, leukocytes subsets are different in the mammary gland and in the milk, a fact that indicates the existence of selective mechanisms of cell transmigration to milk. A recent study compared leukocyte populations in the mammary gland and milk from lactating mice [79]. T cells (mainly CD4^+^ and CD8^+^ T cells with a memory phenotype) constituted the largest leukocyte fraction in milk but were minority in the mammary gland. Inversely, dendritic cells (DCs) represented the largest one in the mammary gland but were minority in milk. In addition, the populations of myeloid and B cells were significantly lower in milk than in mammary tissue. In contrast, another study working with lactating mice found that most leukocytes in the milk bolus were myeloid cells [75]. In relation to CD4^+^ and CD8^+^ T cells, their accumulation in human milk has been reported previously [77,78,80,81,82]. On the basis of animal model studies, it has been speculated that through a high expression of the genes (*Cldn3*, *Cldn7*, *Tjp1*) involved in the biosynthesis of the tight junction (TJ) protein ZO-1 and in the regulation of claudin polymerization [83], CD8^+^ T cells (and maybe CD4^+^ T, γδ T, and NK cells) may selectively translocate through the TJ region of the mammary epithelium while preserving its integrity [79]. As a result, they would reach milk and migrate into neonatal lymphoid tissues [75] to reinforce cell-mediated immunity in the neonatal period [84]. Interestingly, colostrum leukocytes are able to adhere to the gut mucosa of premature baboons, to persist in the wall of the large intestine for, at least, 60 h after a feed, and to reach organs that are particularly interesting in the frame of the development of the immune system, such as the spleen or the liver [85]. This actually represents a mode of maternal microchimerism, and this issue will be discussed below. In contrast, IgA-producing cells are retained inside the mammary gland and this may be regulated by the chemokine CCL28 [86,87]. 

It is generally agreed that, during pregnancy and lactation, leukocytes reach the mammary gland, but the homing to the mammary environment remains an open question, and few works have investigated the attractant and adherence molecules required for their presence there [88]. The T and B cells present in the lactating mammary ecosystem display phenotypes that are quite different to those found in blood and have a homing profile similar to the corresponding cells located in the gut-associated lymphoid tissue (GALT) [75,77,82,89]. The lactating mammary gland is an effector site of the mucosal-associated lymphoid tissue (MALT) system; this implies that following antigens’ exposure in a mucosal inductive site (gastrointestinal or respiratory tracts), lymphocytes may migrate to the unexposed mammary mucosal surface. A previous study reported that the lactating breast compartment is more closely related to the gut mucosa than to that of the upper respiratory tract [82].

Pioneer animal studies revealed that B cells from the maternal gut have the ability to migrate in a specific manner to the mammary gland (the so-called “entero-mammary” pathway or link) during late pregnancy and lactation [87,90,91,92,93,94,95,96]. The homing effect exerted by the mammary gland during such stages is not completely elucidated, but it seems to be under hormonal control [97]; more recent studies have shown the involvement of the interactions between the mucosal vascular addressin MadCAM-1, the gut homing receptor α4β7 integrin [98], and the mucosa-associated CCL28/CCR10 link [99]. Once in the mammary glands, the recruited B cells produce IgAs specifically against the pathogens prevalent in the mother’s environment. In 1975, and while studying the in vitro production of antibodies by human colostral cells after exposure to O antigens from *Escherichia coli*, Ahlstedt et al. [100] suggested that the IgA-forming cells may had been homed to the mammary gland from the gastrointestinal tract after exposure to enteric bacteria. Subsequently, the same research group showed that oral administration of a non-pathogenic *E. coli* strain to pregnant women led to an abundant presence of cells producing IgAs specific against the O antigen of *E. coli* in colostrum samples [101]. Some years later, it was found that the administration of respiratory syncytial virus (RSV) to pregnant rabbits only led to the presence of anti-RSV IgAs in colostrum and milk when viral inoculation was performed by oral or transtracheal routes but not when it was made by an intravenous one [102,103]. The gut-associated IgA system seems to be in a permanent state of adaptation to and protection from the gut microbiota [104]. In fact, the simultaneous vertical transfer of maternal IgA-producing cells and microbes during early life constitutes an elegant mode of inducing a tailored-adaptation of the infant gut to the own mother’s microbiota, fostering the development and maintenance of a healthy gut microbiota, enhancing the barrier function in the enteric environment, exerting immunomodulatory effects, and regulating gene expression of intestinal epithelial cells [105,106,107,108,109]. Studies with lactating mice and their offspring have revealed that the benefits of early exposure to the maternal assembly of IgAs persist into adulthood [89], and may condition allergic or autoimmune disease [110,111]. 

Recently, the existence of an immunologic entero-mammary system modulating the genetic tuning of gut immune responses and inflammatory disease susceptibility has been hypothesized on the basis of murine studies [112]. These authors suggested the existence of a narrow opportunity window (first days after birth) in which the level of bacterial IgA coating in milk inversely correlates with the number of colonic RORγ+ (transcription factors Helios or RORγ) regulatory T cells (Treg) that are induced. In turn, the number of these Treg cells is inversely correlated with the level of IgA in the adult intestine and, as a consequence, with the level of IgA in milk, which would be subsequently transmitted to the next generation. Therefore, this double-negative-feedback loop system would be vertically transmitted, but also serve as a mode of multi-generational matrilineal transmission of immunological information [113]. Intestinal Treg cells play crucial roles in the immune-regulation of host‒pathogen interactions [114] and suppress inadequate innate and adaptive immune responses, promoting tolerance to harmless antigens [113]. The rapid increase in the rates of autoimmune diseases and allergies over the past decades cannot be explained only on the basis of genetic changes [115], and it has been suggested that it may be related to an alteration of maternal immunologic transmission [112].

Similarly to IgA responses to gut microbes, contact with food antigens present in the gastrointestinal tract (such as cow’s milk proteins (CMP)) also direct the specificity of IgAs found in human colostrum and milk [116,117]. These authors showed that high levels of CMP-specific IgAs in human milk conferred protection against cow’s milk allergy, while cow’s milk exclusion from the maternal diet provoked a down-regulation of the levels of such specific IgAs in milk. They also reported that the IgA antibody repertoire towards CMP present in the lactating mammary environment is different to that in the systemic immune compartment, and hypothesized that it could be due to a distinct homing effect [117]. Overall, their results suggest, first, that there is an entero-mammary pathway for food-specific antibody-secreting cells, and second, that elimination diets during pregnancy and breastfeeding may result in decreased levels of protective milk-specific IgAs, and might be related with the development of food allergies in the infant population. 

The finding that DCs are well represented among the leukocyte population of the lactating mammary gland [79] is also interesting in relation to the presence of bacterial cells in milk. DCs are among the professional antigen-presenting cells and play a pivotal role in the regulation and development of innate and adaptive immune responses. After sensing environmental signals (including microbes) from the mucosal surfaces (e.g., gut), they mature, adapt their cytokine microenvironment accordingly, and migrate from the peripheral tissues of the MALT system to secondary lymphoid organs in order to induce and coordinate effector T cell responses. Seminal studies showed DCs and CD18^+^ cells are able to open the tight junctions between gut epithelial cells, uptake non-invasive bacteria from the gut lumen, and transport them to other MALT sites [118,119,120]. The integrity of the epithelial barrier is preserved because of expression of tight junction proteins (ZO-1, occludin, claudin) by the DCs during the penetration of the dendrites [119]. Because of the fact that the lactating mammary gland is a member of the MALT system, Martín et al. [121] hypothesized the existence of an entero-mammary circulation of specific members of the gut microbiota upon complex interactions with gut immune and epithelial cells. Subsequently, it was shown that some lactic acid bacteria strains isolated from human milk had the ability to translocate across a Caco-2 cell monolayer through a DC-mediated mechanism [122,123]. Since then, several in vitro, animal, and human studies have confirmed the plausibility of a physiological bacterial translocation during late pregnancy and lactation [124,125,126,127,128,129,130,131,132,133,134,135,136,137,138,139].

In the last 20 years, several studies have shown that human milk harbors a low biomass microbiota under physiological conditions and that milk, maternal feces, and/or infant feces share some bacteria [24,140,141,142,143]. The ultimate origin (skin, oro-nasopharynx, gastrointestinal tract, vagina) of the microbes present in human milk is difficult to elucidate because of the complex and dynamic interactions between different body locations of the mother‒infant dyad. This is particularly true for the facultative anaerobic bacteria that are usually present in almost any mucosal/epithelial surface of the human body, including those of the digestive, upper respiratory, and genitourinary tracts and/or the skin. They include some of the most abundant bacteria in human milk, such as staphylococci, streptococci, corynebacteria, and cutibacteria. In addition, some gut-associated strict anaerobes have also been isolated or their DNA has been detected in human milk (*Faecalibacterium*, *Roseburia*, *Bifidobacterium*, *Blautia*, *Bacteroides*, *Parabacteroides*, etc.) [24,131,142,144,145]. Their presence is much more difficult to explain on the basis of mere contaminations arising from aerobic body locations, such as the skin or or nasopharynx. Interestingly, the vertical transmission of bifidobacterial communities through breastfeeding also leads to the inheritance of bifidobacterial phages [146]. In fact, bacteriophages represent the vast majority of the human milk viruses and have the ability to modulate the bacterial ecology in the mammary gland and in the infant gut [147]. In relation to the relevance of human milk as a source and/or promoter of *Bifidobacterium* populations in the infant gut, a recent study found that this genus was negatively correlated with the resistome (antibiotic resistance genes) and mobilome (mobile genetic elements) structure, which was shared between human milk and feces within each recruited mother‒infant pair [148]. These authors also found a link between enrichment of resistance genes and antibiotic consumption by mothers, a practice that may have enduring effects on the infant resistome. Similarly, early termination of breastfeeding (which also strongly affects the composition of the infant gut microbiota [149]) was also associated to a resistome enrichment. Therefore, the reduction of antibiotic resistance genes in the infant gut may constitute a new benefit of breastfeeding, especially in the context of the increasing public health threat posed by multi-resistant bacteria.

The potential translocation of intestinal bacteria has generally been related with disease states characterized by alterations in tight junction proteins that favor an increased gut leakiness and the transmigration of microbes and microbial products from the gut lumen into the bloodstream [150,151,152]. However, this process also happens, albeit at a low rate, in healthy hosts and involves the transmigration of mutualistic or beneficial bacteria, including *Bacteroides*, lactobacilli, and bifidobacteria [153,154,155,156,157]. It has been hypothesized that this phenomenon may be linked to physiological immunotraining or immunomodulation of the host [125]. Some studies have suggested that this may be a highly selective process, since some bacterial strains appear to mediate their own translocation without simultaneous translocation of other gut bacteria from the same host [158,159]. Some of the transient body adaptations that take place during pregnancy and lactation are compatible with an intensification of gut permeability and bacterial translocation during such stages [19,24,25]. Recently, Gosalbes et al. [22] investigated how late pregnancy conditions affect gene expression by the gut microbiota and found that during this life stage, some of the bacterial populations seemed to reach a stationary phase. This observation has relevant physiological implications, since growth phase and motility greatly increase the possibility of bacterial translocation [22]. 

Bacterial cells administrated per os may reach locations previously considered sterile environments, including tumors [158,159]. This fact has been explained on the basis of three circumstances that occur in solid tumors: increased nutrient availability (necrotic tumor core), increased vascularization, and suppressed immune surveillance [160,161,162,163,164,165]. It must be highlighted that the similar or identical circumstances also coincide in the mammary gland during late pregnancy: immune-privileged status to tolerate the fetus; intense angiogenesis and vasculogenesis; and a rich nutrient environment, since pre-colostrum is already produced during the last third of pregnancy. These circumstances may also contribute to explain the selective tropism of some maternal bacterial species for the mammary gland during late pregnancy and lactation. More studies are needed to elucidate the mechanisms by which non-pathogenic bacterial strains may translocate in certain life stages. This may provide new strategies for manipulating the infant gut microbiome when it may be required to improve infant health.

## 3. From the Breast to the Infant Gut

### 3.1. Milk: A Multi-Functional Food

Human milk is the best infant feeding option from the nutritional point of view. In addition, this biological fluid contains an impressive array of bioactive components (and it can be anticipated that many others will be discovered in the coming years), making it a very complex and non-imitable functional food. It has become evident that this wealth of nutrients and bioactive ingredients contribute, in a synergic manner, to growth (e.g., insulin-like growth factor (IGF) superfamily, somatostatin, calcitonin), organ development and tissue repair (e.g., stem cells, epidermal growth factor (EGF)), protection against inflammation and infections (e.g., acquired and innate immune factors and cells), enhancement of the mucosal barriers (e.g., SCFAs), immune maturation and tolerance (e.g., transforming growth factor β (TGFβ)), neurodevelopment (e.g., unsaturated fatty acids, neurotransmitters, neuronal growth factors (BDNF, GDNF)), regulation of metabolism and body composition (e.g., adiponectin, ghrelin, leptin), regulation of the vascular system (e.g., vascular endothelial growth factor (VEGF)), prevention of anemia (e.g., erythropoietin), acquisition and development of the microbiota (e.g., oligosaccharides, bacterial cells), and many others. In conclusion, there is no human physiological function that is not, directly or indirectly, influenced by human milk in early life. The complex composition of milk means that it connects with any of the body axes (gut‒brain and gut‒elsewhere axes, hypothalamic–pituitary–adrenal (HPA) axis), intervening in their programming and function. Epidemiological and clinical studies have provided overwhelming evidence that there is a consensus in recognizing that breastfeeding provides relevant short- and long-term benefits for health, both at the individual and at the community levels [16,17].

The contributions of human milk to infant health are the result of interactions among many milk components, and it must be highlighted that most of them (as those cited above) play multifunctional roles. The concentration of many of them is dynamic and, often, adapted to suit infant requirements (age, gestational age at birth, time of the day, environment). In addition, some of them show a high degree of inter-individual variability, and the factors determining their concentrations remain poorly known. Such a complexity makes very difficult, if not impossible, an accurate assessment of the individual and collective roles of human milk components for infant health; in fact, in most cases, this task has remained elusive to scientists, and we are just in the beginning of understanding the functions of individual molecules or cells.

### 3.2. The Complex Case of Human Milk Oligosaccharides

Human milk oligosaccharides (HMOs) are a good example of the multi-functionality of many human milk compounds, but also of the difficulties in unveiling their actual impact on mammary or infant health. At present, approximately 200 different HMOs have been identified, which vary in size, structure, and complexity [166]. They can act as prebiotics, reduce binding of pathogenic microorganisms to the gut cells, and modulate host-epithelial immune responses among other functions, although the effects are often highly structure-specific or depend on the amount and proportion of certain HMOs [167]. Their effects may be not restricted to the gut, since some HMO structures may be absorbed in the gut and, in fact, some of them have already been found in the infant bloodstream [168,169]. The HMO profile of milk varies substantially among women, and the genetic and environmental factors responsible for the inter-individual variability are poorly understood. The peculiar structure of most HMOs makes them difficult to synthesize and/or commercialize from a technical or economical point of view. Therefore, most studies have been restricted to in vitro or animal model experiments, but there is a lack of well-designed and controlled infant studies. In addition, HMOs have mostly been investigated as individual items, when, in contrast, they are present as a diverse and interacting mix in milk and in the infant gut. 

HMOs also interact with other milk compounds. It has been recently found that there is a direct interaction between HMOs and DCs, whereby tolerogenic DCs adopt a regulatory function that induces Treg expansion [170]. In addition, HMOs are able to inhibit LPS-induced pro-inflammatory responses through suppression of LPS-induced maturation of DCs [170]. HMOs selectively stimulate the growth of beneficial bacteria (e.g., bifidobacteria) in the infant gut. As a result of the fermentative activity of these and other bacteria on HMOS, there is an increase of SCFAs in the gut environment. In turn, both bacteria and SCFAs exert a strong influence on DCs phenotype and function [171,172,173,174]. SCFAs and other bacterial metabolites are essential for the cross-talk between the microbiota, the GALT, and the enteric nervous system, while direct interactions between gut DCs and the vagus nerve have also been demonstrated [2]. In relation to the nervous system, some of the bacteria whose growth is promoted by HMOs (including species that have been isolated from human milk) have the ability to participate in the biosynthesis of neurotransmitters (serotonin, dopamine) that have been detected in human milk [175,176,177], and whose presence deserves more research attention. This complex and personalized network of interactions, involving only a few of the human milk ingredients, serves to illustrate that novel combinations of methodologies are required to decipher the multifaceted interactions between them and their collective impact on infant health. 

### 3.3. Role of Human Milk in Immunological Tolerance 

While the importance of breast milk in infectious disease prevention is undisputable, there is uncertainty regarding its role in tolerance induction and allergy prevention [178,179,180,181]. However, recent evidence is accumulating and showing there is a potential of breastmilk for the latter as well that might be critically dependent on the maternal gut‒breast milk‒offspring axis. Two ways by which breast milk can promote tolerance induction in offspring can be distinguished: (a) by promoting immune regulation mechanisms in a non-antigen specific way, and (b) by inducing antigen specific immune tolerance.

In relation to the importance of breast milk for non-antigen specific immune regulation in offspring, and as stated above, breast milk has a major impact on child microbiota seeding and shaping [141,182]. This effect is most probably key in setting immune regulation in offspring, as largely illustrated in observational studies that associate microbiota composition and metabolites production in the first months of life with later allergy risk [183,184,185]. Currently, there are still major gaps of knowledge to be able to exploit the known impact of breast milk on tolerance induction: how to modulate microbiota shaping properties of mother’s milk and what is the ideal microbiota in early life for tolerance in the long term. In addition, and as already exposed in Section 2.2., an elegant study in mice uncovered the mechanisms of transgenerational effect of the maternal gut‒mammary axis on the levels of regulatory Tregs in offspring [112].

Regarding the importance of breast milk in the induction of antigen specific immune tolerance, the absorption of allergens in maternal gut and their shedding in maternal milk might be a key player in dictating allergy risk in offspring. The reports of allergic reactions in exclusively breastfed children demonstrate that the very low amounts of allergens in human milk are sensed by the developing immune system [186]. These observations do not allow a conclusion, however, on whether sensitization occurred through breast milk. Indeed, other sensitizing ways may have occurred, since the consumption of food in a household results in the presence of dietary allergen in dust, which may also lead to sensitization through the infant’s skin [187,188]. A tempting hypothesis to explore is that allergens in human milk might instruct the immune system and induce tolerance. This hypothesis has been widely demonstrated in rodent experimental models [189,190,191,192,193,194,195,196]. Importantly, mechanistic analyses have uncovered that the oral administration of allergen in early life is not sufficient to induce immune tolerance (in contrast to what is observed in adults) and that some maternal milk immune modulators are essential for oral tolerance induction in the developing immune system. Such key milk-derived immune modulators were identified as TGFβ [190], vitamin A [193], IGF-1 [196], and IgG [191,195]. They were found to act at the level of promoting gut barrier and controlling antigen transfer across the gut, antigen presentation by dendritic cells, and induction of regulatory T cells. Today, there is no existing study that formally demonstrates that maternal diet derived allergen influences allergy risk in offspring through its shedding in milk. Only one birth cohort study has analyzed the association between allergen content in human milk itself and allergy risk [197]. Although limited in size, the study showed that having the egg antigen ovalbumin (OVA) in human milk was associated with a four-fold reduction of egg allergy prevalence by the age of 2.5 [197]. Importantly, one cannot conclude on observational studies addressing the association between allergen consumption during lactation and allergy susceptibility in offspring, as multiple reports have shown the unpredictability of allergen content in breast milk. About half of lactating mothers do not secrete allergen in breast milk, even after ingestion of a well-controlled amount of egg, peanut, milk, or wheat [198,199,200]. What is even more surprising is that women consuming an egg exclusion diet were as likely to have detectable egg allergens (OVA and ovomucoid) in breast milk as women with an unmodified healthy diet [200,201,202].

In conclusion, while there is strong experimental support for a role of maternal diet-derived allergens in milk as key for oral tolerance induction in the developing system, much research is needed to elucidate fundamental questions. How is allergen transferred from the gut to the mammary gland? What controls the shedding of diet-derived allergen into maternal milk? Which compartment/configuration in milk make maternal diet-derived antigen the most suitable for tolerance induction? What is the role of micro-vesicles (see Section 3.4) and antigen-presenting cells in these processes? Which immune modulators in human milk are essential co-factors to induce oral tolerance to allergen that are transferred to the neonate through breast milk? How can we modulate those co-factors?

### 3.4. “Emerging” Components of Human Milk: Secreted Extracellular Vesicles and Mirnas

Human milk contains some biologically active entities that have been discovered in the last years. Secreted extracellular vesicles (EVs) are small particles (30–1000 nm) that are produced and exocytosed by many cell types throughout the body. They are packaged with a variety of regulatory molecules from the parent cell, including lipids, proteins, messenger RNA, and microRNAs (miRNAs) that participate in intercellular communication. EVs have attracted extensive research since their discovery in 1983 [203], because the presence of such cargo renders them as therapeutics and diagnosis targets [204]. EVs can be classified into exosomes or microvesicles on the basis of their size [205], although the methodologies currently available do not allow a clear discrimination between them [206]. 

EVs have been identified in milk from many mammalian species [207], including human milk [208,209]. Proteomic analysis of EVs from human milk has revealed the existence of a “novel” functional proteome within these particles, which is distinct to that typically found in milk [210]. In fact, this study identified 633 proteins in human milk-derived EVs that had not been identified in human milk previously, and they included proteins involved in the control of inflammatory signaling pathways and in the regulation of cell growth. The membrane of EVs protects enclosed molecules from digestion and, therefore, they can be delivered intact to the infant gut [211,212]. Gut epithelial cells are able to uptake milk EVs and deliver some of their molecules (e.g., miRNA) to the nucleus. In addition, it has also been suggested that EVs or their content may be transferred into the blood stream, thereby contributing to programming cellular events in distant tissues [209]. 

In vitro studies have demonstrated that human milk EVs promote the growth of intestinal epithelial cells [213] and protect them against oxidative stress [214], while it seems that they display a broad immunomodulatory function, including the activation of Treg cells [208]. The concentration of EVs may change through lactation, since milk collected within the first week after birth has higher concentration of exosomes than milk collected two months after birth [215]. In this context, it is interesting to note that EVs present in preterm and term milk exosomes show different peptide patterns, with an up-regulation of lactotransferrin and lactadherin in EVs from preterm milk, which are responsible for increased rates of cellular growth [216]. This fact has led to the hypothesis that human milk EVs or some of their components may have future applications for the prevention or treatment of necrotizing enterocolitis in preterm neonates [216,217]. 

Most of the research with human milk exosomes has targeted miRNAs, which are small non-coding RNA molecules implied in post-transcriptional regulation of target mRNA. Several abundant miRNAs (e.g., let-7a, let-7b, let-7f, miR-148a) are shared by the milk EVs of different mammalian species [207]. These evolutionarily conserved molecules are involved in regulation of cell growth and immune functions, which are absolutely crucial for the newborn. One study focused on other mammalian species showed that the miRNAs profile of colostrum is more associated with immune regulation, while that of mature milk was more directed to metabolism [218]. In contrast to EVs-associated peptides, the composition of those miRNAs that are highly abundant in human milk EVs is similar in milk from women delivering term and pre-term neonates [211,212]. An alteration of the miRNAs composition may have long-term consequences for health, since it has been shown that the miRNA profile of milk EVs from women with type-1 diabetes is different from that of healthy women, and some of the miRNAs that are differentially-expressed among diabetic women are able to induce the expression of pro-inflammatory cytokines by human monocytes [219]. 

Overall, studies performed so far indicate a relevant role of EVs and miRNAs in the development of the neonatal gastrointestinal tract and immune system and, probably, in other breastfeeding-associated health outcomes [220]. However, similarly to other human milk components, more studies are required to elucidate their roles and their interactions with other milk ingredients. 

### 3.5. The Others That You Carry Inside: Immune Cells, Stem Cells, and Microchimerism 

Different studies have provided evidence supporting the idea that some eukaryotic cells present in human milk may be transferred from the gut mucosa to the tissues of the infant [76,85,221,222,223]. Most of this postnatal maternal microchimerism probably occurs during early lactation when the permeability of both the mammary epithelium and the infant gut epithelium and the concentration of immune and epithelial cells are highest [224]. Little is known about the milk cells that may be involved in microchimerism but, in addition to mature immune cells, they include stem cells. Human milk harbors a whole hierarchy of stem and progenitor stem cells that account for up to 6% of the human cells in human milk and that englobe haemopoietic, mesenchymal, and neuroepithelial lineages [225,226,227]. Because of their properties, human milk stem cells are candidates for microchimerism in the infant tissue [225,228]. Maternal stem cells do not express MHC antigens and, therefore, they can be easily tolerated in infant tissues [224]. In fact, human milk stem cells seem to be able to go across the gut wall of nursed mice and rabbits and reach the bloodstream and, subsequently, different organs where they become functional [229,230,231,232]. More recently, it was found that breast milk stem cells were capable of reaching the brain, where they settled and differentiated into neuronal and glial cell types [233]. Previously, it was found that some stem cells of human milk are able to differentiate in vitro into a variety of cell types, including neurons [227,234].

Independently of the implied cells, breastfeeding-driven maternal microchimerism can complement pregnancy-related bidirectional microchimerism [235,236], contributing to the development of tolerance and to training and maturation of the infant immune system [223], and probably to many other relevant functions that are unknown at present. Given the diversity of cell populations in human milk (from memory lymphocytes and professional antigen-presenting cells to embryonic and mesenchymal stem cells), more research is required to elucidate how milk microchimeric cells may influence health and disease during life.

## 4. Conclusions

From the Industrial Revolution onwards, the rates of women that initiate breastfeeding and, especially, that of infants that are exclusively breastfed for their first six months of life, as recommended by the World Health Organization, started to fall, and the decrease was sharpest from the late 1800s, coinciding with the rapid urbanization of the societies in industrial countries. This trend is also observed in developing countries. However, here we have shown that human milk provides a complex nutritional, immunological, neuroendocrine, and microbiological integration between the mother and the infant, which is specifically tailored to the environment and requirements of the mother‒infant dyad. The axis established first between the maternal gut and breast and, then, between the breast and the infant gut has had, and will have, a paramount role in our existence as a species, serving as a pivotal node interacting with any other human axis and programming health for life. The complexity of the composition of human milk, together with the vast complexity of the interactions among all human milk components, renders this biological fluid a scientific enigma that we are unveiling at a very slow pace. As stated by Bode [166], such complexity “cannot be mimicked in artificial infant formula and provides yet another powerful reason to protect, promote, and support breastfeeding”.

## Figures and Tables

**Figure 1 nutrients-13-00606-f001:**
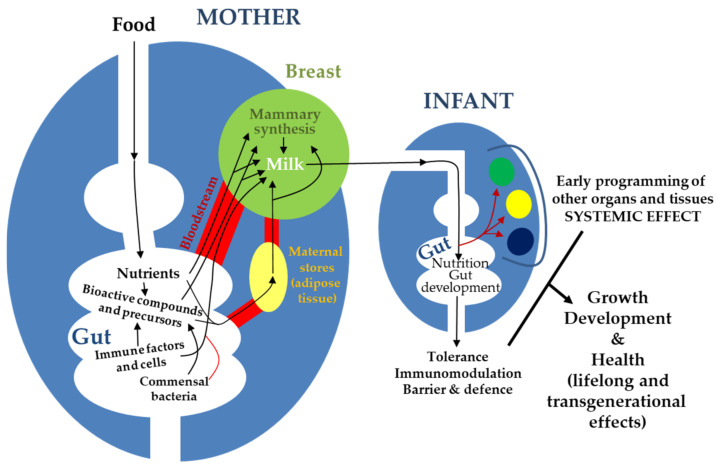
Schematic representation of the complex interactions between maternal gut, milk, and infant gut and their relevance for infant growth, development, and health.

**Figure 2 nutrients-13-00606-f002:**
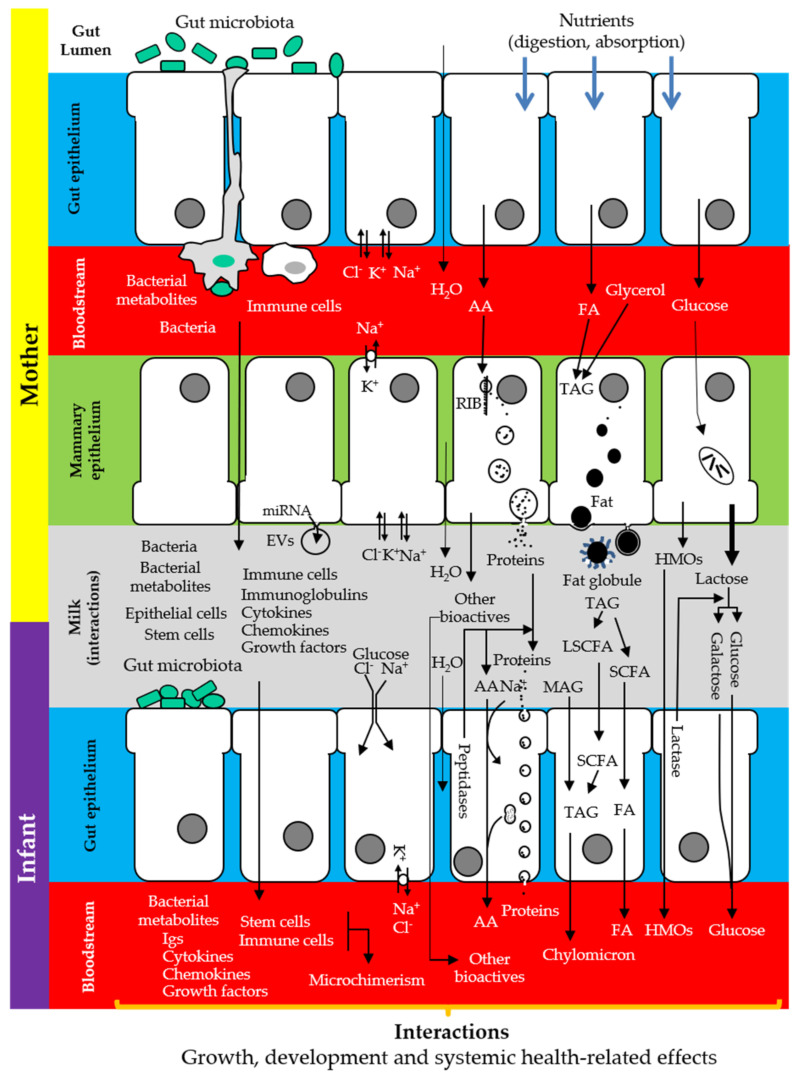
Complementarity between the maternal gut, the breast, and the infant gut, illustrating the synthesis, transfer, use, and/or absorption of some of the components of human milk. AA: amino acids; EVs: extracellular vesicles; FA: fatty acids; HMOs: human milk oligosaccharides; LCFA: long chain fatty acids; MAG: monoacylglycerides; RIB: ribosomes; SCFA: short chain fatty acids; TAG: triacylglycerides. Adapted from [18].

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
