# Peer review of "The Gut‒Breast Axis: Programming Health for Life"

_nutrients, 2021, doi:10.3390/nu13020606_

Round 1
Reviewer 1 Report
The work presented by Rodríguez and collaborators is an excellent review about the complex gut-brain interactions including known evidence and also pointing the gaps and the point where more research is needed. I enjoyed the lecture and I just have very minor suggestions.
Lines 92 and 93: Please consider if these 2 sentences need a reference.
Lines 119-132: Regarding phytochemicals in milk, the sentence starting in line 124 says that polyphenols contribute to the benefits that human milk provides to infants. Are there any epidemiological studies or clinical interventions in humans to prove this? Because for the references it seems just speculative or based on indirect evidence. Please consider.
Author Response
We thank the reviewer for his/her comments. His/her suggestions are actually very helpful for improving our manuscript.
Q1. Lines 92 and 93: Please consider if these 2 sentences need a reference.
Answer: Yes, it seems clear that they need at least a few references. As a consequence, the sentences in the revised manuscript are as follows:
“The concentrations of many of such nutrients are often below recommended levels in many populations. This situation is particularly worrying in low income countries but it can be also observed in developed countries (e.g., omega-3 fatty acids, choline) [27-29, 32-34].”
Two new references (33, 34) have been included:
- Wallace, T.C., Blusztajn, J.K., Caudill, M.A., Klatt, K.C., Natker, E., Zeisel ,S.H., Zelman, K.M. Choline: The underconsumed and underappreciated essential nutrient. Today. 2018, 53, 240-253. doi: 10.1097/NT.0000000000000302.
- Daniels, L., Gibson, R.S., Diana, A., Haszard, J.J., Rahmannia, S., Luftimas, D.E., Hampel, D., Shahab-Ferdows, S., Reid, M., Melo, L., et al. Micronutrient intakes of lactating mothers and their association with breast milk concentrations and micronutrient adequacy of exclusively breastfed Indonesian infants. J. Clin. Nutr. 2019, 110, 391-400. doi: 10.1093/ajcn/nqz047.
Q2. Lines 119-132: Regarding phytochemicals in milk, the sentence starting in line 124 says that polyphenols contribute to the benefits that human milk provides to infants. Are there any epidemiological studies or clinical interventions in humans to prove this? Because for the references it seems just speculative or based on indirect evidence. Please consider.
Answer: Yes, we agree with the reviewer’s point of view. In fact, our intention was to use polyphenols as an example that the health benefits of some components of the diet that may become popular (and lead to a flourishment of food supplements) may not be well established in human studies, yet. We have included a first brief paragraph to talk about endogenous antioxidants in human milk and, then, to cope with the reviewer’s suggestion, we have modified the paragraphs about diet-related polyphenols, as follows:
“Oxidative stress is associated with respiratory and intestinal diseases in infants, especially among preterm ones [54]. Interestingly, human milk contains a wide spectrum of endogenous antioxidant compounds, including superoxide dismutase, glutathione peroxidase, catalase, glutathione or melatonin [55]. The concentration of these antioxidants in human milk is adapted to gestational age, providing higher levels to infants with lower degree of maturation [56]. As a consequence, the level of oxidative stress in premature neonates fed with breast milk is lower than that observed among formula-fed ones [57,58]. Phytochemicals (polyphenols, carotenoids) in the maternal diet provides an exogenous source of antioxidants since they are also transferred to milk [59]. Although many of the human milk phytochemicals have a wide range of biological activities [60-62], most of the research interest has been focused on polyphenols because of their potential as antioxidants. Through their antioxidant activities, diet-related human milk polyphenols might contribute to the benefits that human milk provides to such infant population [63]. Many different flavonoids (epicatechin, epicatechin gallate, epigallocatechin gallate, naringenin, kaempferol, hesperetin, quercetin…) can be present, simultaneously, in the same human milk sample [64]. The flavonoid profile and their concentrations depend almost exclusively on maternal diet and rapid shifts can be observed depending on the intake of polyphenol-rich foods (berries, soya, dark chocolate, olive oil, wine…) [59]. Polyphenol and other antioxidants are becoming increasingly popular as food supplements among the general population but their actual impact in human milk composition and in infant outcomes is rather speculative and far from being fully elucidated because of the lack of well-designed epidemiological studies and human clinical trials.”
We have added some new references about the presence and roles of endogenous antioxidant compounds in human milk:
- Thibeault, D.W. The precarious antioxidant defenses of the preterm infant. Am. J. Perinatol.2000, 17, 167–182. doi: 10.1055/s-2000-9422.
- Gila-Diaz, A., Arribas, S.M., Algara, A., Martín-Cabrejas, M.A., López de Pablo, Á.L., Sáenz de Pipaón, M., Ramiro-Cortijo, D. A review of bioactive factors in human breastmilk: a focus on prematurity. Nutrients. 2019, 11, 1307. doi: 10.3390/nu11061307.
- Gila-Diaz, A., Herranz Carrillo, G., Cañas, S., Saenz de Pipaón, M., Martínez-Orgado, J.A., Rodríguez-Rodríguez, P., López de Pablo, Á.L., Martin-Cabrejas, M.A., Ramiro-Cortijo, D., Arribas, S.M. Influence of maternal age and gestational age on breast milk antioxidants during the first month of lactation. 2020, 12, 2569. doi: 10.3390/nu12092569.
- Friel, J.K.; Martin, S.M.; Langdon, M.; Herzberg, G.R.; Buettner, G.R. Milk from mothers of both premature and full-term infants provides better antioxidant protection than does infant formula. Res.2002, 51, 612–618. doi: 10.1203/00006450-200205000-00012.
- Ledo, A.; Arduini, A.; Asensi, M.A.; Sastre, J.; Escrig, R.; Brugada, M.; Aguar, M.; Saenz, P.; Vento, M. Human milk enhances antioxidant defenses against hydroxyl radical aggression in preterm infants. J. Clin. Nutr.2009, 89, 210–215. doi: 10.3945/ajcn.2008.26845.
Reviewer 2 Report
The review of Rodriguez et al is great and very interesting. It is written understandable and easy to read. The axis explored (gut-breast) and their role in breastmilk production and lactation period is strongly relevant, not only for the industrial proposal but also for clinical reasons. I kindly suggest some references to be considered for the authors to improve the relevant of the review content.
- Line 97-98. Recently it has been found that traza elements in breast milk such as zinc or iron could be below the recommendation intake for infants. These elements also were found higher in breast milk than in plasma of the mothers and also, this context was found in mother with iron supplementation during lactation period. than is interesting to observe the absorption limitation between both elements in de GI system. Maybe, the authors want to explore this field and included it in the text.
- In addition, related to LCPUFAs in breast milk and their contribution on infant gut development, I kindly suggest the reference PMID: 32092925 and the reference PMID: 32854220 for antioxidants in breast milk related to maternal and gestational age.
- Regarding the leukocytes and other cells migration from plasma to the lumen of the mammary gland, it would be interesting to recognize that the present of this type of cells in BM happens more around colostrum stage, when the gland is maturating and the union TJ will be build (PMID: 29221566).
- Related to TJ and breastfeeding there are some article which could be match with this review to be consider for the authors: PMID: 30482190, the breakdown of epithelial barrier due to changes in TJ may explain the transmigration of microbes and microbial products from the gut lumen into the infant blood leading to sepsis. And PMID: 32753526, Maltodextrin-dominant infant formula led to intestinal injury in neonatal mice accompanied by loss of villi, increased MUC2 production, altered expression of TJ, enhanced intestinal permeability, increased cell death and higher levels of intestinal inflammatory mediators.
Minor comments:
- Introduction section should be summarized and focus on why the gut and breast are related on.
- In addition, some figure / squama would reinforce the interesting review.
Author Response
We thank the reviewer for his/her comments. His/her suggestions are actually very helpful for improving our manuscript.
Q1. Line 97-98. Recently it has been found that traza elements in breast milk such as zinc or iron could be below the recommendation intake for infants. These elements also were found higher in breast milk than in plasma of the mothers and also, this context was found in mother with iron supplementation during lactation period. than is interesting to observe the absorption limitation between both elements in de GI system. Maybe, the authors want to explore this field and included it in the text.
Answer: Traza elements in milk is a relevant topic itself and may be the subject of a specific review. However, we agree that it is relevant in the frame of this manuscript and, therefore, we have added a new paragraph in the revised manuscript, as follows:
“Unfortunately, research on how maternal diet affects the micro- and macronutrients’ content of human milk, both quantitatively and qualitatively, is surprisingly scarce. In this context, conflicting data have been obtained in relation to some minerals, such as iron or zinc, which are essential for a wide array of physiological functions [35,36], and which deficiencies in the mother-infant dyad are particularly concerning [37,38]. Some studies have reported that their milk concentrations were associated with maternal dietary intake and/or supplementation, others have found no relationships with these factors and, finally, others have provided ambiguous results [39-43]. More recently, a positive correlation between maternal diet and the milk concentration of both minerals was detected only when the total intake (including diet and supplementation) was taken into account [44]. This and other studies have found that milk levels of these two minerals could be below the adequate intake for infants in a relatively high percentage of samples [34-44]. Positive associations between dietary iron and zinc intakes and their respective serum levels among premenopausal women have been described [45]. However, contrary to iron, zinc concentrations are higher in human milk than in the serum of a breastfeeding woman [46] and, therefore, it is possible that maternal factors may influence the concentration of these two minerals in a different way [44]. Iron and zinc are good examples to highlight that large scale studies are required to provide accurate information about levels, bioavailable forms and critical factors that are associated with the concentrations of micronutrients in human milk.”
We have added new references (related to this new paragraph) in the revised manuscript:
- Eussen, S.; Alles, M.; Uijterschout, L.; Brus, F.; van der Horst-Graat, J. Iron intake and status of children aged 6–36 months in Europe: A systematic review. Nutr. Metab.2015, 66, 80–92. doi: 10.1159/000371357.
- Ackland, L.; Michalczyk, A. Zinc and infant nutrition. Biochem. Biophys.2016, 611, 51–57. doi: 10.1016/j.abb.2016.06.011.
- Petry, N.; Olofin, I.; Boy, E.; Donahue Angel, M.; Rohner, F. The effect of low dose iron and zinc intake on child micronutrient status and development during the first 1000 days of life: a systematic review and meta-Analysis. Nutrients2016, 30, 773. doi: 10.3390/nu8120773.
- Bzikowska-Jura, A.; Czerwonogrodzka-Senczyna, A.; Olędzka, G.; Szostak-Węgierek, D.; Weker, H.; Wesołowska, A. Maternal nutrition and body composition during breastfeeding: association with human milk composition. Nutrients2018, 10, 1379. doi: 10.3390/nu10101379.
- Choi, Y.K.; Kim, J.M.; Lee, J.E.; Cho, M.S.; Kang, B.S.; Choi, H.; Kim, Y. Association of maternal diet with zinc, copper, and iron concentrations in transitional human milk produced by Korean mothers. Nutr. Res.2016, 5, 15–25. doi: 10.7762/cnr.2016.5.1.15.
- Hannan, M.A.; Faraji, B.; Tanguma, J.; Longoria, N.; Rodriguez, R.C. Maternal milk concentration of zinc, iron, selenium, and iodine and its relationship to dietary intakes. Trace Elem. Res.2009, 127, 6–15. doi: 10.1007/s12011-008-8221-9.
- Yalçin, S.S.; Baykan, A.; Yurdakök, K.; Yalçin, S.; GücüÅŸ, A.I. The factors that affect milk-to-serum ratio for iron during early lactation. Pediatr. Hematol. Oncol.2009, 31, 85–90. doi: 10.1097/MPH.0b013e31819146c2.
- Mahdavi, R.; Nikniaz, L.; Gayemmagami, S.J. Association between zinc, copper, and iron concentrations in breast milk and growth of healthy infants in Tabriz, Iran. Trace Elem. Res.2010, 135, 174–181. doi: 10.1007/s12011-009-8510-y.
- Silvestre, M.D.; Lagarda, M.J.; Farre, R.; Martinez-Costa, C.; Brines, J.; Molina, A.; Clemente, G. A study of factors that may influence the determination of copper, iron, and zinc in human milk during sampling and in sample individuals. Trace Elem. Res.2000, 76, 217–227. doi: 10.1385/BTER:76:3:217.
- Bzikowska-Jura, A.; Sobieraj, P.; Michalska-Kacymirow, M.; Wesołowska, A. Investigation of iron and zinc concentrations in human milk in correlation to maternal factors: an observational pilot study in Poland. Nutrients. 2021 Jan 21;13(2):303. doi: 10.3390/nu13020303.
- Lim, K.; Booth, A.; Szymlek-Gay, E.A.; Gibson, R.S.; Bailey, K.B.; Irving, D.; Nowson, C.; Riddell, L. Associations between dietary iron and zinc intakes, and between biochemical iron and zinc status in women. Nutrients2015, 7, 2983-2999. https://doi.org/10.3390/nu7042983
- Nakamori, M.; Ninh, N.X.; Isomura, H.; Yoshiike, N.; Hien, V.T.T.; Nhug, B.T.; Van Nhien, N.; Nakano, T.; Khan, N.C.; Yamamoto, S. Nutritional status of lactating mothers and their breast milk concentration of iron, zinc and copper in rural Vietnam. Nutr. Sci. Vitaminol.2009, 55, 338–345. doi: 10.3177/jnsv.55.338.
Q2. In addition, related to LCPUFAs in breast milk and their contribution on infant gut development, I kindly suggest the reference PMID: 32092925 and the reference PMID: 32854220 for antioxidants in breast milk related to maternal and gestational age.
Answer: Thank you! The reference about LCPUFA is relevant and it has been added in the revised manuscript.
- Ramiro-Cortijo, D.; Singh, P.; Liu, Y.; Medina-Morales, E.; Yakah, W.; Freedman, S.D.; Martin, C.R. Breast milk lipids and fatty acids in regulating neonatal intestinal development and protecting against intestinal injury. 2020, 12, 534. doi: 10.3390/nu12020534.
In addition, we have included a brief paragraph to talk about endogenous antioxidants in human milk, as follows:
“Oxidative stress is associated with respiratory and intestinal diseases in infants, especially among preterm ones [54]. Interestingly, human milk contains a wide spectrum of endogenous antioxidant compounds, including superoxide dismutase, glutathione peroxidase, catalase, glutathione or melatonin [55]. The concentration of these antioxidants in human milk is adapted to gestational age, providing higher levels to infants with lower degree of maturation [56]. As a consequence, the level of oxidative stress in premature neonates fed with breast milk is lower than that observed among formula-fed ones [57,58].”
The following new references (in relation to this paragraph) have been added in the revised manuscript:
- Thibeault, D.W. The precarious antioxidant defenses of the preterm infant. Am. J. Perinatol.2000, 17, 167–182. doi: 10.1055/s-2000-9422.
- Gila-Diaz, A., Arribas, S.M., Algara, A., Martín-Cabrejas, M.A., López de Pablo, Á.L., Sáenz de Pipaón, M., Ramiro-Cortijo, D. A review of bioactive factors in human breastmilk: a focus on prematurity. Nutrients. 2019, 11, 1307. doi: 10.3390/nu11061307.
- Gila-Diaz, A., Herranz Carrillo, G., Cañas, S., Saenz de Pipaón, M., Martínez-Orgado, J.A., Rodríguez-Rodríguez, P., López de Pablo, Á.L., Martin-Cabrejas, M.A., Ramiro-Cortijo, D., Arribas, S.M. Influence of maternal age and gestational age on breast milk antioxidants during the first month of lactation. 2020, 12, 2569. doi: 10.3390/nu12092569.
- Friel, J.K.; Martin, S.M.; Langdon, M.; Herzberg, G.R.; Buettner, G.R. Milk from mothers of both premature and full-term infants provides better antioxidant protection than does infant formula. Res.2002, 51, 612–618. doi: 10.1203/00006450-200205000-00012.
- Ledo, A.; Arduini, A.; Asensi, M.A.; Sastre, J.; Escrig, R.; Brugada, M.; Aguar, M.; Saenz, P.; Vento, M. Human milk enhances antioxidant defenses against hydroxyl radical aggression in preterm infants. J. Clin. Nutr.2009, 89, 210–215. doi: 10.3945/ajcn.2008.26845.
Q3. Regarding the leukocytes and other cells migration from plasma to the lumen of the mammary gland, it would be interesting to recognize that the present of this type of cells in BM happens more around colostrum stage, when the gland is maturating and the union TJ will be build (PMID: 29221566).
Answer: We have introduced an statement including the reviewer’s suggestion in the revised manuscript, as follows:
“The concentration of total leukocytes is higher in colostrum than in mature milk, probably reflecting the fact that tight junctions of the mammary epithelium are looser during the first days after birth [76].”
The suggested reference (PMID: 29221566) deals only marginally with leukocyte migration and, therefore, we have chosen an alternative one:
- Trend, S., de Jong, E., Lloyd, M.L., Kok, C.H., Richmond, P., Doherty, D.A., Simmer, K., Kakulas, F., Strunk, T., Currie, A. Leukocyte populations in human preterm and term breast milk identified by multicolour flow cytometry. PLoS One. 2015, 10, e0135580. doi: 10.1371/journal.pone.0135580.
Q4. Related to TJ and breastfeeding there are some article which could be match with this review to be consider for the authors: PMID: 30482190, the breakdown of epithelial barrier due to changes in TJ may explain the transmigration of microbes and microbial products from the gut lumen into the infant blood leading to sepsis. And PMID: 32753526, Maltodextrin-dominant infant formula led to intestinal injury in neonatal mice accompanied by loss of villi, increased MUC2 production, altered expression of TJ, enhanced intestinal permeability, increased cell death and higher levels of intestinal inflammatory mediators.
Answer: We have introduced an statement including the reviewer’s suggestion in the revised manuscript, as follows:
“The potential translocation of intestinal bacteria has generally been related with disease states characterized by alterations in tight junction proteins that favor an increased gut leakiness and the transmigration of microbes and microbial products from the gut lumen into the bloodstream [150-152].”
We have included one of the suggested references (PMID: 30482190) in the revised manuscript. We think that the second one is only indirectly related to the review subject and we would prefer not to include it. Sorry!
- Ravisankar, S., Tatum, R., Garg, P.M., Herco, M., Shekhawat, P.S., Chen, Y.H. Necrotizing enterocolitis leads to disruption of tight junctions and increase in gut permeability in a mouse model. BMC Pediatr. 2018, 18, 372. doi: 10.1186/s12887-018-1346-x.
Minor comments:
Q5. Introduction section should be summarized and focus on why the gut and breast are related on.
Answer: We appreciate the comment but, if you do not mind, we would prefer to keep the introduction as it was in the original manuscript and unveil why the gut and breast are related in the following sections of the manuscript.
Q6. In addition, some figure / squama would reinforce the interesting review.
Answer: Yes, we agree with the reviewer comment and, as a consequence, we have included two figures in the revised manuscript. We hope you like them.